# Review on Performance of Asphalt and Asphalt Mixture with Waste Cooking Oil

**DOI:** 10.3390/ma16041341

**Published:** 2023-02-04

**Authors:** Yaofei Luo, Ke Zhang

**Affiliations:** 1School of Civil Engineering and Architecture, Zhengzhou University of Aeronautics, Zhengzhou 450046, China; 2College of Information Engineering, Fuyang Normal University, Fuyang 236041, China

**Keywords:** waste cooking oil, modified asphalt, asphalt recycling agent, road performance, aged asphalt

## Abstract

To make full use of the regenerative value of waste cooking oil, and to solve the environmental pollution and food security issues caused by waste cooking oil, waste cooking oil was suggested for use in asphalt. Waste cooking oil was used to adjust the performance of virgin and aged asphalt. This review article summarizes research progress on the performance of asphalt and asphalt mixture with waste cooking oil. The results showed that a moderate dosage of waste cooking oil will improved the low-temperature performance and construction workability of petroleum asphalt and aged asphalt. The mixing and compaction temperature of asphalt mixture with waste cooking oil are reduced by up to 15 °C. The rutting resistance and fatigue resistance of modified asphalt and modified asphalt mixture with waste cooking oil are damaged. After the addition of waste cooking oil in aged asphalt, the high-temperature performance and shear rheologic property of aged asphalt will be recovered. The regeneration effect of waste cooking oil on aged asphalt and aged asphalt mixture is close to that of a traditional regeneration agent, and the partial performance of asphalt or asphalt mixture with waste cooking oil is better. There is no chemical reaction between waste cooking oil and asphalt, but the asphalt component and absorption peak intensity of partial functional groups are changed. The light components content of asphalt binder is usually increased. Further research regarding the engineering application of asphalt mixture with waste cooking oil should be conducted. The method for improving the performance of asphalt and asphalt mixture with waste cooking oil will be mainly researched.

## 1. Introduction

The waste cooking oil is generated from the cooking and food production process. The waste edible vegetable oil and waste edible animal oil are classified as waste cooking oil. The waste cooking oil is mainly included the waste oil after repeatedly fried food, recycled waste oil from catering trade, gutter oil and condensate oil in smoke lampblack machine. According to the statistical results from the China National Grain & Oils Information Center, the human consumption of edible oil was 3545.0 million tonnes in 2020 in China, and an average of 500~700 million tonnes of waste cooking oil is produced every year. There are a lot of pathogenetic components and carcinogenic components in waste cooking oil, and the dosage of heavy metal, aflatoxin and benzopyrene exceeds the limit of standard scope. Once the waste cooking oil is flowed back to people’s dining table, the food security and human health will be seriously threatened [1]. Currently, the waste cooking oil is usually in the form of commercial grease, stearic acid, industrial oleic acid, chemical raw material, biodiesel, release agent, modifier and regenerating agent [2,3]. The waste cooking oil can be used in road engineering, and it is not decolored and deacidified. The waste cooking oil can also reduce the viscosity of a binder, and it may not produce secondary pollution [4].

The main component of waste cooking oil is fatty acid, which belongs to aromatic oil, and it is similar to the aromatic fraction in asphalt components. So, the waste cooking oil can be used to soften asphalt, restore the aged asphalt performance, and regulate the asphalt components and rheological property [5,6,7]. The compatibility between asphalt and modifier is improved by the addition of waste cooking oil, and the partial performance of asphalt with waste cooking oil can be adjusted. The petroleum-based rejuvenators are widely used to restore the aged asphalt performance, but there are still many technical problems [8,9]: (1) the light components of petroleum-based rejuvenators are easily volatilized at high temperature, so the regeneration efficiency of petroleum-based rejuvenators is relatively poor; (2) the content of aromatics and unsaturated bonds of petroleum-based rejuvenators is relatively high; thus, rejuvenators are oxidized easily at high temperature. The aging resistance and durability of petroleum-based rejuvenators are reduced. Meanwhile, the petroleum-based rejuvenators are originated from petroleum, which consumes non-renewable resources; thus, it cannot meet the direction of sustainable development.

The waste cooking oil is used as an asphalt-regenerating agent, which can realize the recycling of waste asphalt pavement materials and waste cooking oil. The partial performance of recycled asphalt with waste cooking oil is better than that with other waste oil. The asphalt regenerant prepared by waste cooking oil, waste bio oil and waste engine oil is added into 70# aged asphalt, when the dosage of asphalt regenerant is less than 13%, the penetration and softening point of recycled asphalt with waste cooking oil is higher than that of other recycled asphalt, but the ductility of recycled asphalt with waste cooking oil is the worst [10]. The different regenerant with the best dosage is prepared rejuvenated asphalt, the colloidal stability of recycled asphalt with waste cooking oil is inferior to waste bio oil and waste engine oil, and the rejuvenated asphalt with waste cooking oil is drastically aged [11]. The improvement effect of waste engine oil on the low-temperature performance of aged asphalt is less than that of waste vegetable oil. When the dosage of different regenerant is the same in the rejuvenated asphalt mixture, the strength of the rejuvenated asphalt mixture with waste engine oil is higher, but the durability of the rejuvenated asphalt mixture with waste cooking oil is much better [12].

However, the recoverability rate of waste cooking oil is relatively low in China, and the waste cooking oil is a potential environmental pollution source. If not be treated properly, it will give rise to soil pollution and environmental pollution. To increase the recycling utilization rate of waste cooking oil in the asphalt pavement, the application of waste cooking oil used as asphalt modifier, asphalt regenerant and biological binder is discussed basing on the domestic and abroad research. Articles discussing “waste cooking oil and modified asphalt” or “waste cooking oil and aged asphalt” or “waste cooking oil and asphalt mixture” were searched for in Web of Science and www.cnki.net. The published articles from 2010 to 2022 were chosen and analyzed in this paper, especially published articles in the last five years. Only the articles form core journals were included in the above published articles.

The ordinary performance and rheological properties of binder are analyzed, and the road performance of asphalt mixture is explained. The main factors influencing the performance of modified asphalt and asphalt mixture with waste cooking oil are discussed. The microscopic test, scanning electron microscopy and four fractions separation test are selected to measure the mechanism between waste cooking oil and asphalt binder, and the reaction between waste cooking oil and asphalt binder is illustrated. The existential problem in the current research and application field of waste cooking oil in asphalt pavement is proposed, which will provide reference for the widely application of waste cooking oil in asphalt pavement.

## 2. Research on the Performance and Mechanism of Asphalt and Asphalt Mixture with Waste Cooking Oil

### 2.1. Influence of Waste Cooking Oil on the Performance of Asphalt and Asphalt Mixture

#### 2.1.1. Research on Performance of Modified Asphalt with Waste Cooking Oil

The waste cooking oil is added into petroleum asphalt by the shearing process, after which the viscosity, stiffness, elastic recovery performance, temperature sensitivity, rutting resistance and resistance to fatigue cracking of modified asphalt with waste cooking oil are reduced. The above performance of modified asphalt with waste cooking oil is gradually declined with increasing the dosage of waste cooking oil. However, the low-temperature crack resistance, fatigue resistance and self-healing efficiency of modified asphalt with waste cooking oil are improved.

Wen et al. [13] has studied that the PG grades of other blend modified asphalt and found that they are reduced except the blend modified asphalt with PG76-22 base asphalt and the dosage of 10% waste cooking oil. The resistance to fatigue cracking of modified asphalt with waste cooking oil is reduced, which is decreased with the increase in the amount of waste cooking oil. Azahar et al. [14] has pointed out that the modified asphalt performance is significantly affected by the quality of waste cooking oil, which is attributed to the interaction bonding between the waste cooking oil and asphalt binder. After the waste cooking oil is chemically treated, the acid value reduces from 1.66 to 0.54 mL/g. Due to the increase in interaction bonding between the treated waste cooking oil particles and the particles in the asphalt binder, the softening point, viscosity, rutting resistance and aging index of blended asphalt with the above waste cooking oil are improved, but the penetration and temperature sensitivity of blended asphalt are reduced. Eriskin et al. [15] found that the softening point of modified asphalt with waste frying oil is decreased, but the penetration value is increased. So, the modified asphalt with waste frying oil is suggested to be used in a colder region. Wang et al. [16] has evaluated the aging performance of different asphalt binder by the four fractions separation test, gel permeation chromatography and thermo-gravimetric analysis. It is found that the dispersed system of modified asphalt is improved by the addition of bio-oil produced from waste cooking oil, and the bio-oil is used to balance the effect of aging. The aging performance of bio-oil modified asphalt prepared by waste cooking oil is inferior to petroleum asphalt, but the thermal stability of the above asphalt is essentially the same. The colloidal index and molecular weight of bio-oil modified asphalt are not increased with increasing the dosage of the waste cooking oil. The different dosage of waste cooking oil (5%, 10% and 15%) is added into 70# base asphalt and SBS-modified asphalt, and the failure stress of modified asphalt with waste cooking oil is reduced, but the failure strain and fatigue life at an intermediate temperature of modified asphalt with waste cooking oil are increased. In addition, the yield energy of modified asphalt with waste cooking oil is gradually declined upon increasing the amount of the waste cooking oil [17].

Sun et al. [18] has studied that the bio-oil form by-product of waste cooking oil and found that it has good compatibility with petroleum asphalt through the separation tendency test of polymer-modified asphalt; the softening point differences of different blended asphalt are less than 2.5 °C. The content of asphaltene in bio-oil is less than 1%, but the dosage of saturates and resins is higher than that of base asphalt. Maharaj and Singh-Ackbarali et al. [19,20] have found that the influence of waste cooking oil on the asphalt elasticity is closely related to the chemical composition of binder. Upon increasing the dosage of waste cooking oil, the phase angle of rock asphalt-modified asphalt with waste cooking oil is gradually increased, the phase angle of blended asphalt with waste cooking oil and petroleum asphalt is declined, and the phase angle of composite-modified asphalt with waste cooking oil is first decreased and then increased, which is shown in Figure 1. Ma et al. [21] has studied the preparation process, and the dosage of waste cooking oil residue of rubber asphalt modified with waste cooking oil residue are determined by an orthogonal design method, and the best preparation process is sheared for 2 h at 220 °C with 6.0% waste cooking oil residue. The storage stability of rubber asphalt is improved by the addition of waste cooking oil residue, which can help to promote the swelling and degradation of rubber particles (as shown in Figure 2), and the aging resistance of rubber asphalt with waste cooking oil residue is also improved. The interaction between crumb rubber and asphalt is reduced by the addition of waste cooking oil residue, and the waste cooking oil residue can increase the proportion of lubricant phase. Gökalp et al. [22] found that the waste vegetable cooking oil significantly decreases the softening point, viscosity and rutting factor of asphalt. The waste vegetable cooking oil can be utilized as an anti-aging agent of aged asphalt. Liu et al. [23] has pointed out that the storage stability of SBS/EVA-modified asphalt is improved by the addition of waste cooking oil, and the compatibility of the SBS and EVA is better when the dosage of waste cooking oil is 10%. Due to waste cooking oil being mainly composed by low-weight components, the light components of polymer-modified asphalt with waste cooking oil are supplemented, the content of medium molecular size is increased (as listed in Table 1), and the internal microscopic structure of asphalt becomes more smooth. Li et al. [24] has found that the optimal dosage of waste cooking oil in epoxy asphalt binder is determined as 4% by the consideration of viscosity, microstructure, damping behaviors and mechanical performance. The viscosity of the binder and particle size of the dispersed phase is reduced, the damping behavior and elongation at break are improved, and the construction time of the modified asphalt is extended to be long as 24%. The influence of waste cooking oil on the performance of asphalt is closely related to the base asphalt as well as the dosage and quality of waste cooking oil.

Considering the overall performance of the asphalt binder, the dosage of waste cooking oil added into petroleum asphalt is generally lower than 10%. When the dosage of waste cooking oil is too high, the performance of the asphalt binder is damaged. Sun et al. [25,26] prepared the bio-asphalt with high content waste cooking oil residues and discussed the influence of different modifiers on the bio-asphalt performance. The study pointed out that the improvement effect of composite modifier with rock asphalt, hydrocarbon resin, low-density polyethylene and linear SBS polymer on bio-asphalt performance is relatively the best, and the optimum dosage of different modifiers is determined by the uniform design method. The high-temperature performance of the above bio-asphalt with a composite modifier is close to SBS-modified asphalt, the low-temperature performance of the above bio-asphalt is better, and the aging sensitivity of the above bio-asphalt is inferior to that of the 70# base asphalt.

The physical property of partial waste cooking oil is listed in Table 2, and the other waste cooking oil is collected directly from restaurants.

To achieve the obvious improvement effect of waste cooking oil on asphalt binder performance, the waste cooking oil should satisfy the following performance requirements [16,17,18,21,23,27]: (1) The waste cooking oil has good fluidity, the viscosity range of waste cooking oil should be 100~200 mPa·s at 25 °C; (2) The density difference between waste cooking oil and asphalt could not too large, and the density of waste cooking oil should be greater than 0.90 g/cm^3^ at °C; (3) The dosage of impurity in waste cooking oil is low; (4) The moisture content of waste cooking oil should be low; otherwise, the security of asphalt with waste cooking oil cannot be guaranteed.

#### 2.1.2. Research on Performance of Modified Asphalt Mixture with Waste Cooking Oil

A mixture of modified asphalt with bio-oil derived from waste cooking oil or bio-asphalt derived from waste cooking oil is prepared, and the construction temperature of asphalt mixture is also decreased due to the reduction in viscosity of the binder [27]. The construction temperature of asphalt mixture with waste cooking oil is listed in Table 3. The dynamic modulus, rut resistance and fatigue resistance of asphalt mixture with waste cooking oil are decreased with increasing the amount of the waste cooking oil, the low-temperature crack resistance of asphalt mixture with waste cooking oil is gradually improved, but there is no obvious correlation between the moisture susceptibility of asphalt mixture with waste cooking oil and the dosage of waste cooking oil. Meanwhile, the tensile strength ratio of asphalt mixture with the different dosages of bio-asphalt produced from waste cooking oil can all meet the requirements [13]. The density of asphalt mixture with untreated waste cooking oil or treated waste cooking oil is increased, which is because that the waste cooking oil has a lubricating function and good flowability. The frictional force between the aggregate and aggregate is reduced by the addition of waste cooking oil, so the asphalt mixture with waste cooking oil is easy to compact [31].

Eriskin et al. [15] has found that the optimum asphalt content, indirect tensile strength and self-healing temperature of modified bitumen mixture with waste flying oil are declined with the increasing dosage of waste flying oil, and the decrease in optimum asphalt content is up to 89%. There is a decline in the tensile strength ratio of modified bitumen mixture with the increasing dosage of waste flying oil, and the tensile strength ratio of the above mixtures can meet specification requirements. Azahar et al. [32] has studied the compactness, elastic modulus, indirect tensile strength and creep resistance of modified asphalt mixture with treated waste cooking oil and found that they are higher than the base asphalt mixture and modified asphalt mixture with untreated waste cooking oil, and the highest creep stiffness of modified asphalt mixture with treated waste cooking oil is improved about 25% compared with the base asphalt mixture. Niu et al. [33] has found that the Marshall stability, immersion Marshall residual, dynamic stability and tensile strength ratio of modified asphalt mixture with 5% waste cooking oil are all lower than those of the original asphalt mixture, but the performance of the modified asphalt mixture with 5% waste cooking oil is improved by the addition of ground tire rubber. The modified asphalt mixture with 5% waste cooking oil and 20% ground tire rubber has better rut resistance, water stability and low-temperature performance. Yan et al. [34] has evaluated the mechanical behaviors of asphalt mixture with Europan rock and waste cooking oil, and they found that the anti-cracking performance of asphalt mixture at low temperature can be improved by addition of waste cooking oil. Due to the addition of Europan rock, the anti-cracking performance of asphalt mixture is reduced, but the waste cooking oil made up for the adverse effect of Europan rock on the anti-cracking performance of the mixture. The different performance of asphalt mixture is significantly improved by adding the proper dosage of Europan rock and waste cooking oil. Hu et al. [35] has studied the noise reduction performance of porous asphalt mixture with Sasobit and waste cooking oil. The workability of high-viscosity asphalt rubber is improved by the addition of Sasobit and waste cooking oil, and the viscosity of high-viscosity asphalt rubber with waste cooking oil is lower. The damping performance of porous asphalt rubber mixture waste cooking oil is improved, so the vibration noise and friction noise of the above mixture is smaller.

After the addition of waste cooking oil into an asphalt mixture, the temperature stability and strength of the asphalt mixture are markedly affected, but the influence of waste cooking oil on the moisture susceptibility of the asphalt mixture is relatively small. The partial performance of the asphalt mixture with waste cooking oil is damaged, and the waste cooking oil is not recommended to be added alone into asphalt mixture. To ensure the maximum performance of the asphalt mixture, the waste cooking oil and other modifiers (rubber powder, SBS and SBR) are blended into asphalt binder [33,34].

### 2.2. Function Mechanism between Waste Cooking Oil and Asphalt Binder

The function mechanism between waste cooking oil and petroleum asphalt or polymer is measured by a microscopic test. The physical blend between waste cooking oil and base asphalt occurs though a transform infrared spectroscopy test, and the almost no chemical reaction happens. The new absorption peaks appear in the modified asphalt with dosages of 4% and 8% waste cooking oil, but the above absorption peak position is corresponding with the infrared spectrum of waste cooking oil, and it shows that no new absorption peak has appeared [18]. The infrared spectroscopy test result is listed in Table 4. Wang et al. [17] found that the functional groups of 70# base asphalt after the addition of waste cooking oil are nearly unchanged, and the sulfoxide index remains almost the same, but the carbonyl index is increased, which is listed in Table 5. Research confirms that there is no chemical reaction between waste cooking oil or waste cooking oil residue and polymer-modified asphalt, such as rubber-modified asphalt, SBS/EVA composite-modified asphalt and SBS-modified asphalt, but there are some differences in the absorption peak intensity of only a few functional groups [21,23,27]. Sun et al. [27] has found that specific functional groups (functional groups of carbonyl, carbon–oxygen band, methylene) have approximately linear relationships with the dosage of bio-oil from waste cooking oil (listed in Table 6), which means that the bio-oil and SBS-modified bitumen are mainly physically mixed. Liu et al. [23] has pointed out that the carbonyl index and sulfoxide index of SBS/EVA-modified asphalt are observed to be generally increased by the addition of waste cooking oil (listed in Table 7), and the absorption intensity of the peaks has changed, which may be the reason for the change of asphalt performance.

## 3. Study on the Performance and Mechanism of Rejuvenated Asphalt Binder and Mixture with Waste Cooking Oil

### 3.1. Research on the Performance of Rejuvenated Asphalt Binder and Mixture with Waste Cooking Oil

#### 3.1.1. Performance of Rejuvenated-Asphalt Binder with Waste Cooking Oil

The waste cooking oil as a low-viscosity material is used to regenerate aged asphalt, and the low-temperature crack resistance, fatigue performance, adhesion and construction workability of aged asphalt are improved. The ductility of aged asphalt is increased by the addition of waste cooking oil, but the rotational viscosity of rejuvenated asphalt binder with waste cooking oil is decreased.

Yan et al. [36] found that the low-temperature performance of rejuvenated asphalt binder with 8% fried food waste oil is equivalent to that of virgin asphalt binder. The mixing temperature and paving temperature of rejuvenated asphalt binder with fried food waste oil are decreased. The study pointed out that the ductility and viscosity of aged asphalt are recovered by the addition of waste cooking oil, and the above indicators of aged asphalt with 3% waste cooking oil are close to those of the base asphalt level [37], as listed in Table 8. Wan et al. [38] pointed out that the flexibility at low temperature and crack resistance of rejuvenated-asphalt binder with waste cooking oil are improved, and the improvement effect of the above asphalt is better than that of waste lubricating oil, which is listed in Table 9. Yang et al. [39] found that the waste cooking oil and WROB modified rejuvenator with waste tire crumb rubber and waste cooking oil are conducive to the low-temperature anti-cracking performance of aged asphalt, but the WROB-modified rejuvenator has superior performance in recovering the low-temperature crack resistance ability of aged asphalt, as listed in Table 10. Uz et al. [40] pointed out that viscosity of aged asphalt with 6% waste vegetable cooking oils is close to that of base asphalt. Li et al. [41,42] found that the waste cooking oil has a significant influence on the viscosity of recycled asphalt, and the viscosity of recycled asphalt is decreased with the increase in dosage of waste cooking oil. The study points that the ductility and viscosity of rejuvenated asphalt with waste cooking oil can be recovered to the level of base asphalt, which is related to the dosage and macromolecular substances of waste cooking oil [43,44]. Bilema and Sun et al. [45,46] pointed out that the ductility of aged asphalt is increased and the viscosity of aged asphalt is decreased by addition of waste cooking oil, which is closely related to type and dosage of waste cooking oil. Ma et al. [47] found that the asphalt with waste bio-oil has a better low-temperature performance, and the high-temperature performance of asphalt with waste bio-oil is improved by the addition of Iran rock asphalt. Bilema et al. [48] pointed out that the workability of reclaimed asphalt is improved by the addition of waste frying oil and crumb rubber. Ren et al. [49] found that the addition of waste cooking oil remarkably can restore the viscosity and strengthen the low-temperature performance of rejuvenated asphalt. However, the other performance of rejuvenated asphalt is damaged, so the waste cooking oil and styrene butadiene rubber are used to recover performance of aged asphalt. The study points out that the viscosity of the aged asphalt with waste cooking oil is more effectively restored, and the viscosity of aged asphalt with a reasonable dose of waste cooking oil can be reduced by 60.3% and 52.5% [50].

Zhao et al. [51,52] pointed out that the modification effect of waste cooking oil on low-temperature performance is related to the components of waste cooking oil, and the light components of waste cooking oil have a limited recovery effect on the low-temperature ductility of aging asphalt. Lai et al. [53] found that the addition of waste cooking oil can reduce the risk of the material cracking under extremely cold conditions, but the ductility of rejuvenated asphalt binder is increased at first and then decreased with increasing the amount of waste cooking oil. When the waste edible vegetable oil is, respectively, added into the SBS-modified asphalt, AH-70#asphalt and AH-50# asphalt, there is a decrease in the viscosity of different aged asphalt (aged SBS-modified asphalt, aged AH-50# asphalt and aged AH-70#asphalt), and there is an increase in the ductility of different asphalt (aged AH-70#asphalt, aged AH-50# asphalt and aged SBS-modified asphalt) [4,54]. Zhang et al. [55] pointed out that the performance of rejuvenated asphalt is affected markedly by the quality of waste cooking oil, and especially the aging performance and crack resistance of rejuvenated asphalt are closely related to the acid value and viscosity of waste cooking oil. The acid value and viscosity of waste cooking oil are lower the better the regeneration effect of aged asphalt is. The study found with the increase in waste oil contents, the viscosity of aged asphalt is gradually decreased. The waste cooking oil has the most obvious effect on viscosity reduction, and the viscosity of aged asphalt with the highest dosage of waste cooking oil is 54.4% lower than that of aged asphalt [56]. The partial test results are listed in Table 11.

The construction safety, high-temperature stability and anti-aging property of aged asphalt are reduced by the addition of waste cooking oil. The performance of rejuvenated asphalt binder is closely related to the type of aged asphalt and waste cooking oil, aging degree of binder and waste oil. The influence of waste cooking oil on the performance of aged asphalt is summarized in Table 11. The flash point of rejuvenated asphalt with different dosages of waste cooking oil is decreased, which shows that the waste cooking oil is harmful to the safety of aged asphalt. However, the flash point of rejuvenated asphalt with 20% waste cooking oil or 3% waste vegetable oils is still higher than 230 °C [62,64].

The waste edible vegetable oil is added into different asphalt, AH-70# asphalt and AH-50# asphalt after spending time in a pressurized aging vessel, after which the increase in penetration for aged AH-70# asphalt, aged SBS-modified asphalt and aged AH-50# asphalt is gradually decreased. In addition, the influence of waste edible vegetable oil on the thermo-physical properties of the above aged asphalt is evidently different [54]. Zheng [59] found that the regeneration effect of aged asphalt is significantly influenced by the aging degree of waste soybean oil. When the aged soybean oil with a viscosity is less than 1792 mPa.s is added into the aged asphalt, the lost light component of aged asphalt can be effectively supplemented. Ji et al. [60] studied the influence of different rejuvenators with waste cooking vegetable oil on aged asphalt performance and found that the influence of waste corn oil on the penetration and softening point is lower than waste soybean oil, and the high temperature performance of recovered asphalt binder with waste corn oil are better than waste soybean oil. Li et al. [68] found that there is a difference in the regeneration effect of waste soybean oil on the asphalt with different aging degrees; when the dosage of 1~3% waste soybean oil is added into 70# asphalt with the aging time for 5 h, 7 h, 9 h, 11 h, 13 h and 15 h, the penetration of the above asphalt is recovered to 102.9%, 87.3%, 93.4%, 98.1%, 107.7% and 102.9% of the base asphalt. After the addition of 1~2% of waste soybean oil into asphalt aged for 5 h, 7 h, 9 h, 11 h, 13 h and 15 h, the softening point of the above asphalt is recovered up to 90.7%, 95.1%, 98.1%, 97.5%, 103.1% and 94.9%. Ji et al. [66] found that the influence of ultraviolet aging on the fatigue resistance and elastic recovery performance of the 70#-rejuvenated asphalt is significantly higher than that of the SBS-rejuvenated asphalt. The three different types of waste fried oil with a mass ratio of 4~12% are blended into 70# aged asphalt, and the influence effect of the selected waste oil on the recovered asphalt binder performance is different. The improvement effect of the selected waste oil on the high-temperature performance of the recovered asphalt binder is: waste cooking oil ② > waste cooking oil ③ > waste cooking oil ① [69], which is listed in Table 12. Matolia et al. [70] pointed out that the high-temperature performance and viscosity of different aged asphalt are reduced by the addition of waste vegetable oil, and the recovery effect is relate to the aging degree of asphalt and dosage of waste vegetable oil. The surface free energy of aged asphalt with waste vegetable oil is improved. The dosage of 2~10% waste vegetable cooking oil is added into the asphalt after the short-term aging and the long-term aging, the changing range of penetration for rejuvenated asphalt after the short-term aging is higher than the rejuvenated asphalt after the long-term aging, but the softening point of rejuvenated asphalt after the long-term aging has a greater change [40]. The study points out that waste pig fat has good compatibility with aged asphalt, and the durability of aged asphalt is increased by the addition of waste pig fat [71]. Yan et al. [72,73] found that the adhesion of asphalt and aggregate is decreased due to the aging process, but the surface free energy of aged asphalt is improved by the addition of waste cooking oil. The adhesion work of asphalt and selected aggregate is different, which is listed in Table 13.

The dosage of waste cooking oil in rejuvenated asphalt does not adhere to the concept: “the more the better”. The high-temperature performance and shear rheologic property of rejuvenated asphalt with an appropriate dosage of waste cooking oil can be equal to base asphalt, but the conclusion is not uniform whether the viscosity and low-temperature performance of the above rejuvenated asphalt can be recovered to the base asphalt level. There is a difference in the optimum dosage of waste cooking oil in aged asphalt determined by the conventional performance indexes or rheological property indexes, as listed in Table 14, Table 15 and Table 16, which may be related to the test method, grade and aging degree of asphalt binder, and the type, quality and viscosity of waste cooking oil. Man [74] found that the regeneration effect of a waste vegetable oil regenerator on aged asphalt is slightly inferior to that of a traditional rejuvenating agent, but the optimum dosage of the waste vegetable oil regenerator is significantly lower when the aged asphalt performance is recovered.

#### 3.1.2. Research on the Performance of Waste Cooking Oil Rejuvenated Asphalt Mixture

The low-temperature crack resistance of the aged asphalt mixture is improved with the increasing of waste cooking oil content. The maximum bending strain of the aged asphalt mixture with waste cooking oil contents of 0, 4%, 8% and 12% is 2633.32 µε, 2879.25 µε, 3087.44 µε and 3246.13µε [53]. Yan et al. [76] found that after the addition of waste cooking oil into an aged asphalt mixture, the indirect tensile strength is increased, but the failure strain is decreased.

With the increasing dosage of waste cooking oil, the void ratio, Marshall stability, flow value, indirect tensile strength and fatigue life of a rejuvenated asphalt mixture with waste cooking oil are reduced. The permanent deformation resistance, freeze–thaw splitting strength ratio and residual stability of the above asphalt mixture are first increased and then decreased, and the moisture susceptibility of the above asphalt mixture is evidently improved [63,74,77,78]. Sun et al. [67] found that the Marshall stability of waste vegetable oil rejuvenated asphalt mixture is slightly lower than that of the base asphalt mixture, but the flow value of waste vegetable oil rejuvenated asphalt mixture is higher by about 35%. Yan et al. [76] found that the temperature stability, moisture susceptibility and anti-aging performance of a rejuvenated asphalt mixture with tung oil are better than those with waste cooking oil, but the fatigue performance of the mixture with waste cooking oil is better. The tensile strength ratio and Marshall stability of an asphalt mixture are decreased significantly after accelerated aging, and the moisture susceptibility cannot meet specification requirements. The moisture susceptibility of an aged asphalt mixture is improved by the addition of 12% waste cooking oil, but the Marshall stability still could not meet specification requirements [78]. Ziari et al. [79] pointed out that the rutting resistance of a rejuvenated mixture with waste cooking oil or waste engine oil is decreased, which is attributed to the lubricant effect of waste cooking oil and waste engine oil. To rectify the negative effect of waste cooking oil or waste engine oil on the rutting resistance of a rejuvenated mixture, 25% crumb rubber is blended with waste cooking oil and waste engine oil to produce modified waste cooking oil and modified waste engine oil. The viscosity of modified waste cooking oil and modified waste engine oil is increased, so the rutting resistance of the rejuvenated mixture with modified waste oil is improved.

When the performance of a rejuvenated asphalt mixture with waste oil can meet specification requirements, the dosage of waste cooking oil in rejuvenated asphalt mixture is generally lower than that of waste engine oil. The density, Marshall stability, flow value, indirect tensile strength and freeze–thaw splitting strength ratio of a rejuvenated asphalt mixture with waste cooking oil are better than those with waste engine oil [63,78]. However, the dosage of waste cooking oil is the same as that in the rejuvenated asphalt mixture with waste engine oil, and the permanent deformation of the waste engine oil rejuvenated asphalt mixture is lower [78]. Compared to the rejuvenated asphalt mixture with a commercial rejuvenating agent, the performance of a waste cooking oil-rejuvenated asphalt mixture is not necessarily better, which is listed in Table 16.

**Table 16 materials-16-01341-t016:** Performance and strength of different rejuvenated asphalt.

Researcher	Performance or Strength of Different Rejuvenated Asphalt
Man [80]	High-temperature performance and moisture susceptibility: waste cooking oil > RPO rejuvenated asphalt mixture, low-temperature performance: waste cooking oil < RPO rejuvenated asphalt mixture
Hassan Ziari et al. [81]	Rutting resistance: waste cooking oil > cyclogen and Rapiol rejuvenated asphalt mixture, low-temperature performance and moisture susceptibility: closely approximate
Mamun et al. [82]	Indirect tensile strength and resilience modulus: waste cooking oil > SAE-10 rejuvenated asphalt mixture, loss rate of indirect tensile strength: little difference

The aged asphalt performance is closely related to the physical properties of waste cooking oil. To achieve good performance of rejuvenated-asphalt binder and mixture with waste cooking oil, the waste cooking oil used as rejuvenating agent should satisfy the following technical requirements [4,54,62,64,65]: (1) The viscosity of waste cooking oil is no more than 1.8 Pa.s at 25 °C, which can finely dispersed in aged asphalt; (2) The density of waste cooking oil is 0.90~1.0 g/cm^3^ at 15 °C, which can obtain better compatibility between aged asphalt and waste cooking oil; (3) Good aging resistance, heat stability and weather resistance; (4) Clean without impurities, and the impurities should be removed.

### 3.2. Analysis on Function Mechanism between Waste Cooking Oil and Aged Asphalt

There is no chemical reaction between waste cooking oil and aged asphalt. The high polar sulfoxide group of aged asphalt is diluted by the addition of waste cooking oil, and the dosage of macromolecules and small molecules in aged asphalt are reduced, which helps to improve the dispersion of molecular weight for aged asphalt [55,58,61,64,65,68]. Meanwhile, the four components of aged asphalt are also changed by the addition of a rejuvenating agent obtained by waste cooking oil, but its colloidal structure is not changed. Zhang and Matolia et.al [55,70] have found that the content of asphaltene and resin of rejuvenated asphalt are reduced compared to that of the aged asphalt, and the saturates fraction and aromatic fraction of rejuvenated asphalt are increased. However, some researchers have pointed out that the content of saturated fraction and resin of regenerated asphalt are increased, and the content of asphaltene and aromatic fraction of regenerated asphalt are declined [54,68], which is listed in Table 17. The structure of regenerated asphalt is the same as that of the aged asphalt, and their main non-electrochemistry displacement is basically similar, but the chemical electrical property of regenerated asphalt is only changed. The electrochemical displacement of aged asphalt and regenerated asphalt is −56~22 ppm and 10~60 ppm, and the light components are increased by the waste cooking oil [67].

The functional group of aged asphalt has a high similarity with rejuvenated asphalt with waste cooking oil though the infrared spectroscopy test, but the intensity of functional groups for aged asphalt at about 1600 cm^−1^ and 1030 cm^−1^ is different from that of the rejuvenated asphalt with waste cooking oil. This shows that the intensity of the above functional groups of rejuvenated asphalt with waste cooking oil is decreased markedly [58,64,68], as is listed in Table 18 and Table 19. Adding waste cooking oil into the aged asphalt, the ratio of asphaltenes to maltenes can be reduced but cannot restored. The microscopic image of aged asphalt and rejuvenated asphalt with waste cooking oil is observed by the scanning electron microscopy (SEM), and the researchers found that the spread bores with different sizes on the surface morphology of the waste cooking oil rejuvenated asphalt disappeared, and the morphological characteristics of waste cooking oil rejuvenated asphalt became relatively smooth [63], which is shown in Figure 3. However, Li et al. [68] found that the micro-morphology of aged asphalt and rejuvenated asphalt has no remarkably difference, and some little tiny white points appeared in the rejuvenated asphalt surface. The microcosmic appearance of rejuvenated asphalt is apparently no longer homogeneous, which is shown in Figure 4. Compared to aged asphalt, the components composition of rejuvenated asphalt has changed. The property changes of asphalt are related to the components composition. The resins and saturated content of rejuvenated asphalt are increased, so the ductility of asphalt is improved and viscosity is decreased. The softening point, viscosity and ductility of aged asphalt are recovered by the addition of waste cooking oil, which shows that the frictional resistance and low-temperature performance of rejuvenated asphalt are improved.

## 4. Discussion

In this paper, the waste cooking oil can be used as a modifier and regenerating agent applied in asphalt binder. The influence of waste cooking oil on asphalt and asphalt mixture performance is discussed, and the influence factor of asphalt with waste cooking oil is also analyzed. The performance of asphalt with waste cooking oil is related to the asphalt type, content and physical property of waste cooking oil. The low-temperature crack resistance, fatigue resistance, workability and self-healing efficiency of asphalt with waste cooking oil are improved, but the high-temperature performance of asphalt with waste cooking oil is reduced. The performance changes of asphalt mixture with waste cooking oil are similar to those of asphalt with waste cooking oil. Methods for improving the performance of asphalt and asphalt mixture with waste cooking oil are suggested as areas of further study, which can increase the applied range of asphalt with waste cooking oil.

The regeneration efficiency of waste cooking oil used in aged asphalt is good. The performance of rejuvenated asphalt with waste cooking oil is nearly comparable to that of petroleum-based rejuvenator. The consumption of petroleum-based rejuvenator is reduced, which can accord with sustainable development. More waste cooking oil is encouraged to be collected and used to recover the performance of aged asphalt. Upon consideration of the influence of waste cooking oil on rejuvenated asphalt and asphalt mixture performance, the waste cooking oil should satisfy the technical requirements, such as viscosity, density and cleanliness.

The method of distinguishing quality for waste cooking oil is proposed. After the waste cooking oil is collected, the waste cooking oil is distilled to remove moisture. The physical indicators of treated waste cooking oil are tested, and it is blended with virgin asphalt or aged asphalt according to the same preparation process. Based on the basic performance and compatibility of asphalt with waste cooking oil, such as the low-temperature performance, viscosity, adhesion and high-temperature performance, the waste cooking oil can be classified into different levels.

## 5. Further Research Interests

The application feasibility of WCO into asphalt and asphalt mixture was analyzed in the above study, and the influence of WCO on the performance of asphalt and asphalt mixture was discussed, which was conducive to reveal the mechanism of WCO and asphalt binder. The following aspects will need to be further researched.

The type and quality of WCO are not mentioned in the above study. Soybean oil, peanut oil and rapeseed oil are usually eaten, but the type and aging degree of WCO from the above oils are not explicitly discussed. This is may be the important reason for the performance differences of different rejuvenated asphalt and blends composed by WCO and binder, and the standardized application of WCO into asphalt and asphalt mixture will not be realized. Therefore, the influence of type and quality of WCO on asphalt and asphalt mixture performance will be studied; the performance difference of selected WCO is analyzed, and the quantitative quality control parameters of WCO will be proposed.

The research on asphalt and asphalt mixture with WCO is still in a laboratorial testing stage at present, and the asphalt and asphalt mixture with WCO have not been applied in road projects. The influence of WCO on asphalt and asphalt mixture performance has not been uniform, and the modified asphalt with WCO or WCO rejuvenated asphalt binder performance may not work under the coupling effects of natural environment and vehicle load. Therefore, the improvement of composite technology on modified asphalt with WCO or WCO rejuvenated asphalt binder performance will be suggested to be conducted, which will improve the overall performance of the above asphalt, and the economy of different technologies will be also considered. Ultimately, the WCO will be realized with resourceful utilization into asphalt pavement.

## 6. Conclusions

The low-temperature crack resistance and construction workability of modified asphalt with waste cooking oil are improved. However, the stiffness, elasticity recovery, temperature sensitivity, rutting resistance and resistance to fatigue cracking of modified asphalt with waste cooking oil are reduced. The type of base asphalt, dosage and quality of waste cooking oil have a significant influence on the modified asphalt performance. The dosage of waste cooking oil added into petroleum asphalt is generally lower than 10%. The compatibility between base asphalt and different polymer is improved by the addition of waste cooking oil, the storage stability of polymer-modified asphalt with waste cooking oil is enhanced, and the light components of polymer-modified asphalt are increased.

The construction temperature, optimum asphalt content, indirect tensile strength, dynamic modulus, rut resistance and fatigue resistance of rejuvenated asphalt mixture with waste cooking oil are reduced, but the low-temperature crack resistance of rejuvenated asphalt mixture with waste cooking oil is improved. The construction temperature of waste cooking oil rejuvenated asphalt mixture is reduced by up to 15 °C, and the moisture susceptibility of rejuvenated asphalt mixture is less influenced by waste cooking oil.

Performance of the rejuvenated asphalt mixture with waste cooking oil is closely related to the type and dosage of waste cooking oil, aged asphalt type, and aging degree of asphalt and waste cooking oil. However, the dosage of waste cooking oil in aged asphalt is not the more the better, and the optimum content of waste cooking oil is 1.0~15.0%.The high-temperature performance and shear rheologic property of aged asphalt with an appropriate dosage of waste cooking oil are recovered.

The waste cooking oil is blended with base asphalt or aged asphalt; there is no chemical reaction between waste cooking oil and asphalt. However, there are differences in the absorption peak intensity of the individual functional group in modified asphalt with waste cooking oil. Meanwhile the structure of rejuvenated asphalt with waste cooking oil is the same as that of the aged asphalt, the asphaltene components of select asphalt are changed, and the content of saturated fraction is increased.

## Figures and Tables

**Figure 1 materials-16-01341-f001:**
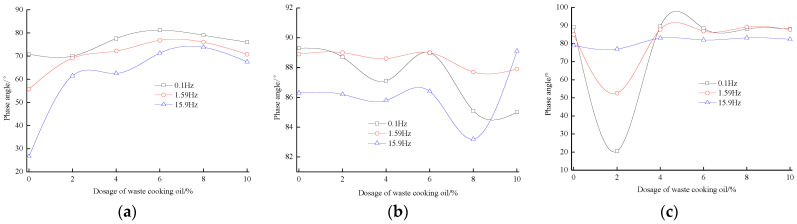
The interrelation between the phase angle and dosage of modified asphalt with waste cooking oil [19]. (**a**) Rock asphalt. (**b**) Petroleum asphalt. (**c**) Rock asphalt/Petroleum asphalt.

**Figure 2 materials-16-01341-f002:**
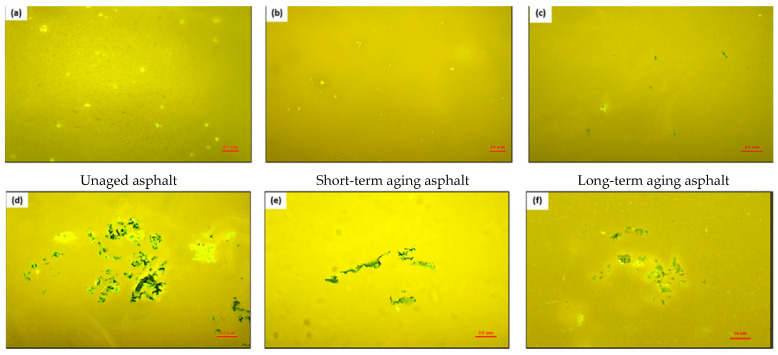
Segregation test result of different blended asphalt: (**a**) virgin blended asphalt with WCO and rubber asphalt; (**b**) short-term aged blended asphalt with WCO and rubber asphalt; (**c**) long-term aged blended asphalt with WCO and rubber asphalt; (**d**) virgin rubber asphalt; (**e**) short-term aged rubber asphalt; (**f**) long-term aged rubber asphalt [21].

**Figure 3 materials-16-01341-f003:**
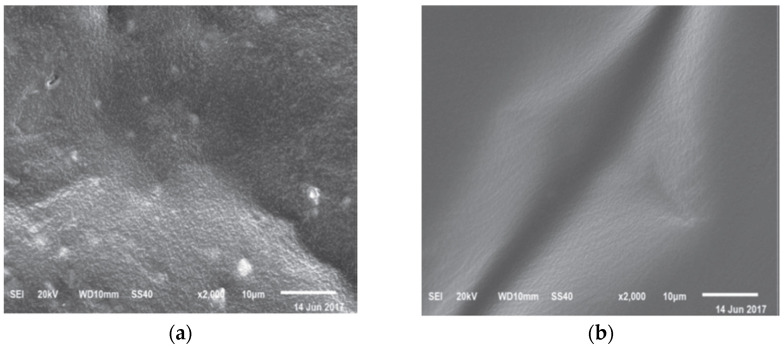
SEM image of aged and rejuvenated-asphalt [63]. (**a**) Aged asphalt, (**b**) rejuvenated-asphalt.

**Figure 4 materials-16-01341-f004:**
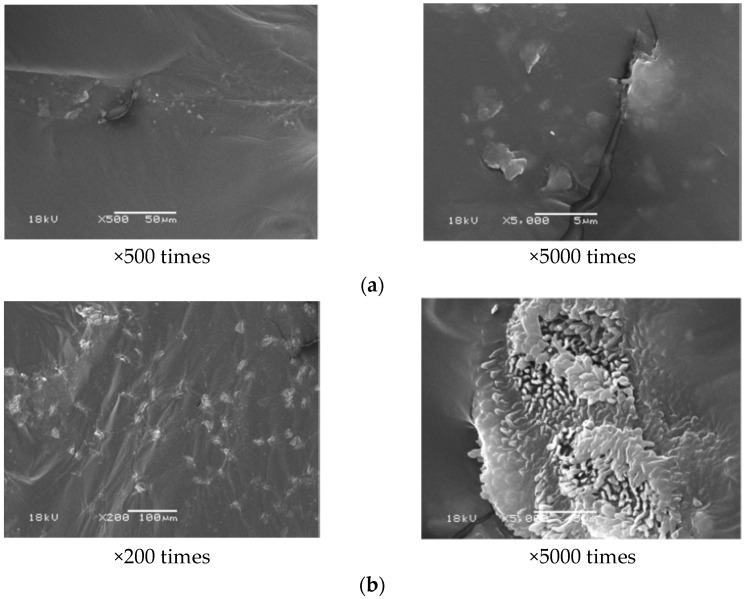
Microscopic image of aged and rejuvenated-asphalt [68]. (**a**) Aged asphalt, (**b**) rejuvenated asphalt.

**Table 1 materials-16-01341-t001:** Analysis on chemical composition based on gel permeation chromatography [23].

Asphalt Type	Content of Large Molecular Size/%	Content of Medium Molecular Size/%	Content of Small Molecular Size/%
Base asphalt	11.74	81.64	6.62
SBS/EVA-modified asphalt	17.39	74.15	8.46
SBS/EVA composite-modified asphalt with 5% waste cooking oil	16.97	74.92	8.11
SBS/EVA composite-modified asphalt with 10% waste cooking oil	15.58	76.38	8.04

**Table 2 materials-16-01341-t002:** Physical property of some waste cooking oil.

Material	Source	Physical Indicators	Literature
Bio-asphalt	Produced from waste cooking oil after undergoing the thermochemical process	Specific gravity:1.54.	[13]
Bio-oil	Waste cooking oil recycled from restaurants	Appearance: Black, sticky and non-smoky liquid; Density: 0.90 g/cm^3^ at 15 °C; Rotational viscosity: 139.5 mPa s at 25 °C.	[16,17]
Bio-oil	By-product of waste cooking oil refining for biodiesel	Appearance: black oily liquid; Moisture content: 3.1%; Density: 0.95 g/cm^3^ at 15 °C; Rotational viscosity: 146.3 mPa s at 25 °C.	[18,27]
Waste cooking oil residue	The by-product of biodiesel refined using waste cooking oil feedstock	Density: 0.95 g/cm^3^ at room temperature; Acid value: 45~55 mgKOH/g.	[21]
Waste cooking oil	A commercial restaurant	Density: 0.913 g/cm^3^ at 20 °C; Water content: 0.18%; Acid value: 15.4 mgKOH/g.	[23]
Waste cooling oil	A fried food factory	Density: 0.93 g/cm^3^ at 15 °C; Viscosity: 0.18 Pa s at 25 °C; Mass change after TFOT: 0.50%.	[28]
Waste oil	Produced by industrial steps such as high-temperature cracking of waste biological oil	Density: 0.92 g/cm^3^ at 15 °C; Viscosity: 88.6 Pa s at 60 °C; Acid value: 55.8 mgKOH/g.	[29]
Waste cooling oil	Refining process for bio-diesel production	Density: 0.915 g/cm^3^; Viscosity: 0.15 Pa s at 60 °C; pH: 7.3.	[30]

**Table 3 materials-16-01341-t003:** Mixing temperature and compaction temperature of SBS-modified asphalt with waste cooking oil [27].

Dosage of Waste Cooking Oil/%	Mixing Temperature/°C			Compaction Temperature/°C		
	Lower	Upper	Average	Lower	Upper	Average
0	169.4	176.3	172.9	156.2	161.8	159.0
4	163.4	170.5	167.0	150.1	155.8	153.0
8	156.2	163.2	159.7	143.1	148.7	145.9
12	148.1	154.4	151.3	136.0	141.1	138.6
16	141.3	147.5	144.4	129.5	134.5	132.0

**Table 4 materials-16-01341-t004:** Wave number and functional group of absorption peak [18].

Position	Material Type				Functional Group
	Waste Cooking Oil	Base Asphalt	Modified Asphalt with 4% Waste Cooking Oil	Modified Asphalt with 8% Waste Cooking Oil	
Absorption peak wavenumbers/cm^−1^	727	741	741	741	-(CH_2_)_n_-
	—	760	752	757	C_6_H_5_-
	—	820	817	820	
	—	884	878	882	
	—	1058	1038	1055	S = O
	1181	—	1188	1190	-CO-O-
	1380	1379	1379	1379	C-CH_3_
	1467	1465	1465	1464	
	—	1616	1611	1617	C_6_H_5_-
	1748	—	1744	1745	-CO-O-
	2861	2859	2874	2859	-CH_2_-
	2945	2933	2943	2932	
	3486	3482	3486	3483	-CO-NH-

**Table 5 materials-16-01341-t005:** Calculation results of sulfoxide index and carbonyl index [17].

Material Type	*I* _C = O_	*I* _S = O_
70# base asphalt	0.031	0.033
70# base asphalt + 5%waste cooking oil	/	0.032
70# base asphalt + 10% waste cooking oil	0.039	0.031
70# base asphalt + 15% waste cooking oil	0.042	0.032
waste cooking oil	0.068	0.047

**Table 6 materials-16-01341-t006:** Correlations between functional group indices and bio-oil content [27].

Bio-Oil Content/%	*I* _C = O_	*I* _C-O_	*I* _CH2_
0	0.13	0.12	0.35
8	0.80	0.30	0.41
16	1.18	0.51	0.48

**Table 7 materials-16-01341-t007:** Results of sulfoxide index and carbonyl index [21].

Types	Waste Cooking Oil	Base Asphalt	SBS/EVA-Modified Asphalt	SBS/EVA-Modified Asphalt with 5% Waste Cooking Oil	SBS/EVA-Modified Asphalt with 10% Waste Cooking Oil
Carbonyl index	0.072	0.046	0.035	0.046	0.052
Sulfoxide index	0.094	0.077	0.026	0.067	0.085

**Table 8 materials-16-01341-t008:** Viscosity and ductility of different asphalt [37].

Types	Ductility at 10 °C/cm	Viscosity at 135 °C/Pa.s
Base asphalt	33.1	0.503
Aged asphalt	8.2	0.727
Aged asphalt with 2% waste cooking oil	18.3	0.613
Aged asphalt with 3% waste cooking oil	31.5	0.532
Aged asphalt with 4% waste cooking oil	43.4	0.462

**Table 9 materials-16-01341-t009:** Influence of different waste oil on ductility of aged asphalt [38].

Dosage of Waste Oil	Waste Cooking Oil					Waste Lubricating Oil				
	0	3	5	7	9	0	5	10	15	20
Ductility at 15 °C/cm	57	190	500	>500	>500	57	124	214	280	446

**Table 10 materials-16-01341-t010:** High and low-temperature performance grade of different asphalt binder [39].

Types	High-Temperature PG/°C	Low-Temperature PG/°C
Base asphalt	64	−22
Aged asphalt	82	−16
Aged asphalt with WROB60	76	−28
Aged asphalt with WROB80	70	−22
Aged asphalt with WROB100	70	−22
Aged asphalt with Evoflex8182	76	−22

**Table 11 materials-16-01341-t011:** Influence of waste cooking oil on performance of aged asphalt.

Researcher	Original Asphalt	Type and Dosage of Waste Cooking Oil	Change of Performance and Strength
Asli et al. [57]	Penetration grade 80/100	Waste cooking oil, 1~5%	Softening point and viscosity of aged asphalt are reduced with the increase in waste cooking oil content, and the softening point of aged asphalt with 4% waste cooking oil can reach the base asphalt level.
Majid Zargar et al. [58]	Penetration grade 40/50	Waste cooking oil, 1~5%	Penetration is increased and change range of penetration is 53~132 (0.1 mm), softening point is reduced and change range of softening point is 48~41 °C, viscosity is reduced and change range of viscosity is 490~285 mPa.s.
Zheng [59]	AH-90# asphalt	Waste soybean oil, 6%	Softening point of aged asphalt with waste soybean oil is deceased and ductility is increased. The recover effect of aged asphalt performance is related to the aging degree of soybean oil.
Ji et al. [60]	PG 64-22 asphalt binder	Waste cooking vegetable oils, 2%~10%	The high-temperature performance and viscosity of aged asphalt with waste cooking vegetable oils are reduced with the increase in waste cooking vegetable oils content, and the ductility is increased. The recovering efficiency of aged asphalt is related to the type and dosage of waste oil.
Cao et al. [61]	50# asphalt	Waste vegetable oil, 5%,10%,15% and 20%. Viscosity of waste vegetable oil: 286.7 mPa.s at 60 °C; Density: 0.966 g/cm^3^	Viscosity is respectively reduced by 4~19.6%, 5~17.7%, 2~17.6% at 90, 135 and 180 °C, the linear viscoelastic range of maximum amplitude is increased by 6.1%, 7.0%, 10.1% and 13.3%.
Cao et al. [62]	50# asphalt	Waste vegetable oil,5%,10%,15% and 20%. Viscosity: 286.7 mPa.s at 60 °C; Density: 0.966 g/cm^3^	Flash point and softening point are reduced by 7~16.3% and 11.3~36.0%, the penetration is increased by 85~355%, the ductility is improved by 4.6~25 times.
Amira et al. [63]	Penetration grade 60/70	Waste cooking oil, 2%, 3%, 3.5% and 4%	Penetration is increased by about 47~95%, softening point is reduced by about 15~29%, viscosity is respectively reduced by about 55~60.5%, 41~55%, 57~61.6% at 135, 150 and 165 °C.
Hasan et al. [64]	Penetration grade 20/30	Waste cooking oil, 1%~3%	Penetration and ductility is increased by 86~254% and 47~78.2%, the softening point, viscosity and flash point are reduced by 2~13%, 19.4~29.0% and 4~6.7%.
Li et al. [61]	70# asphalt	Waste soybean oil, 1%~5%	The high-temperature performance and viscosity of aged asphalt with waste cooking vegetable oils are decreased with the increase in soybean oil content, and ductility is increased. The recovering efficiency of aged asphalt is related to the aging time of asphalt and dosage of waste oil.
Tang et al. [65]	Donghai-50# asphalt	Waste oil from purification of soybean oil, 5%, 10%, 15% and 20%	Penetration is increased and the change range of penetration is 38~104 (0.1 mm), softening point is reduced and change range of softening point is 58.8~45.5 °C, and ductility is increased and change range of viscosity is 5.8~35 cm.
Ji et al. [66]	70# asphalt	Waste soybean oil, 4%	The fatigue resistance and elastic recovery performance of 70# rejuvenated asphalt with waste soybean oil are reduced after ultraviolet aging.
Sun et al. [67]	Shell-70# asphalt	Frying food oil, 5%, 10%, 15% and 20%	Penetration is increased by 1.42~2.77 time, softening point is reduced by about 10.4~32.6%, and ductility is improved.

**Table 12 materials-16-01341-t012:** Physical property of new oil and waste cooking oil [69].

Material	Moisture Content after Treatment/%	Density at 15 °C/g.cm^−3^	Viscosity at 25 °C/Pa.s	Flash Point/°C
New oil	<0.01	0.921	0.05	240
Waste cooking oil ①	0.01	0.909	0.06	242
Waste cooking oil ②	0.09	0.914	0.07	251
Waste cooking oil ③	0.06	0.912	0.07	246

**Table 13 materials-16-01341-t013:** Adhesion work between asphalt and aggregate [72].

Asphalt Types	Adhesion Work/mJ.m^2^			
	Basalt	Limestone	Sandstone	Granite
Virgin asphalt	103.6	79.0	74.5	73.2
Aged asphalt	70.2	53.4	49.2	46.3
Aged asphalt + 2% waste cooking oil	89.6	67.9	64.0	61.7
Aged asphalt + 4% waste cooking oil	102.3	77.7	73.6	72.0
Aged asphalt + 6% waste cooking oil	118.0	89.4	85.4	84.3
Aged asphalt + 8% waste cooking oil	128.3	97.1	93.1	92.5
Aged asphalt + 10% waste cooking oil	134.2	101.7	97.7	97.4

**Table 14 materials-16-01341-t014:** Optimum content of waste cooking oil in aged asphalt.

Researcher	Original Asphalt	Type and Physical Index of Waste Cooking Oil	Optimum Content	Reference Index
Leng et al. [4]	SBS-modified asphalt, AH-70# and AH-50# asphalt	Viscosity: 0.05 Pa.s at 20 °C; Density: 0.896 g/cm^3^ at 25 °C.	4%, 5% and 6%	Penetration, softening point, ductility and viscosity
Chen et al. [54]	Penetration grade of base asphalt is 60~80, 40~60 and 40~60 (A0, B0), and SBS modified asphalt (C0)	Viscosity of frying vegetable oil: 0.05 Pa.s at 25 °C; Density: 0.896 g/cm^3^ at 25 °C.	6.6%, 5.2% and 4.8%	Penetration, softening point, ductility and viscosity
Majid Zargar et al. [58]	Penetration grade 80/100	Waste vegetable oil	3~4%	Penetration, softening point, ductility, complex shear modulus and phase angle
Cao et al. [62]	70# asphalt	Viscosity of by-product after the extraction of fatty acid from vegetable oil. Viscosity: 286.7 mPa.s at 60 °C; Density: 0.966 g/cm^3^.	13.4%	Non-recoverable creep compliance
Amira M. El-Shorbagy et al. [63]	Penetration grade 60/70	Waste cooking oil	3.5%	Penetration, softening point and viscosity
Hasan H. Joni et al. [64]	Penetration grade 40/50	Viscosity of waste vegetable oil: 156 cp; Specific gravity: 0.92; Water content: 0.31%.	1.0%	Penetration, softening point and ductility
Tang et al. [65]	Donghai-50#asphalt	Viscosity of waste oil from purification of soybean oil: 287 mPa.s sat 60 °C; Density: 0.966 g/cm^3^; Flash point: 276 °C; Mass loss: 0.8%.	10%	Penetration, softening point and ductility
Sun et al. [67]	Shell-70# asphalt	Frying food oil	15%	Penetration, softening point and ductility
Cao et al. [75]	Donghai AH-70#asphalt	Frying soybean oil	4.5%	Penetration

**Table 15 materials-16-01341-t015:** Optimum content of waste oil based on determining different physical properties [70].

Aging Degree of Asphalt	Determining Optimum Content of Waste Oil Based on Viscosity at 60 °C/%	Determining Optimum Content of Waste Oil Based on Softening Point at 60 °C/%
Short-term aging for 85 min	5.5	6.0
Short-term aging for 5 h	7.0	7.5
Long-term aging for 20 h	8.5	10.0
Long-term aging for 48 h	10.0	10.0
Long-term aging for 72 h	12.0	12.0

**Table 17 materials-16-01341-t017:** Components of aged asphalt and rejuvenated-asphalt.

Researcher	Asphalt Type	Content of Different Components/%			
		Asphaltene	Resin	Saturates Fraction	Aromatic Fraction
Matolia et al. [70]	Aged asphalt	28.0	38.3	11.2	22.5
	Aged asphalt + 10% waste vegetable oil	18.7	35.2	13.5	32.6
Li et al. [68]	Aged asphalt	17.46	26.33	4.25	51.96
	Aged asphalt + 2% waste cooking oil	15.64	31.57	6.22	46.57

**Table 18 materials-16-01341-t018:** Absorbance of key functional group [58].

Material	Absorbance	
	C = O Carbonyl	S = O Sulfoxide Compound
Virgin asphalt	0.027	0.005
Aged asphalt	-	0.011
Rejuvenated asphalt	0.032	0.009
Waste cooking oil	0.026	0.005

**Table 19 materials-16-01341-t019:** Intensity o of key functional group [68].

Material	Transmittance/%	
	C = O Carbonyl	S = O Sulfoxide
Aged asphalt	80.7	93.4
Aged asphalt with 1% waste cooking oil	83.8	92.3
Aged asphalt with 2% waste cooking oil	88.0	93.4
Aged asphalt with 3% waste cooking oil	85.9	93.4

## Data Availability

The data presented in this study are available on request from the corresponding author.

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
