# Peer review of "Review on Performance of Asphalt and Asphalt Mixture with Waste Cooking Oil"

_materials, 2023, doi:10.3390/ma16041341_

Round 1
Reviewer 1 Report
This paper reviews the performance of asphalt modified and rejuvenated with waste cooking oil. The manuscript should be revised with more summary of current researches and more Figures/Tables should be provided to give us a more readable paper rather than lots of descriptions. The detailed comments are listed below for author's consideration.
- my main criticism is that this paper not focusing on the objectives; rather, we read that low-temperature crack was discussed and then Authors discussing the rutting performance, The current paper is very confusing. My suggestion is that authors should rewrite the sections (2) by focusing first on the low-temperature performance for asphalt and asphalt mixtures, then the following subsection will be on high-temperature performance of asphalt and asphalt mix, lastly, authors can discuss the chemical interaction of asphalt by referring to SARA fractions analysis, FTIR test, and other test.
- The number of references is not sufficient for review paper ( should be about 100 references)and also most of the cite references is outdated. Pls cite more recent and relevant paper (2018 onward).
- Authors should revise the title by including phrase of review in the title.
- Abstract is very informative and lengthy, Pls summarize it by including the main aim, methods, main findings (Pros and cons) and further studies.
- Pls change the number of introduction to 1.
-Pls cite reference for text from line 46 to 53, also for line 69. sentence from line 78 to 85.
- Authors are comparing the WCO with waste engine oil in the abstract and other part of the study; what is the aim of this comparison?
- it is stated " The conventional tests and rheological tests of asphalt and asphalt mixture are selected to analyze the influence of waste cooking oil on the pavement performance of asphalt and asphalt mixture" What do authors mean by he conventional tests and rheological tests of asphalt and asphalt mixture; i think there are no such test for asphalt mixture. The conventional tests and rheological tests are used for asphalt binder only.
- Pls cite the proper references for Table 1, Table 2, Fig 1, Fig 2, Fig3, Table 3, Table 4, Table 5-11, and fig 5.
- use Table for comparing the main findings for functional group.
- Pls revise the sentence from line 329 to 333.
- Pls put the number of reference after"Man" in line 357.
- Pls refer to proper references for technical requirements for waste cooking oil (line 415 to 418)
- pls revise the sentence "and the influence of waste cooking oil on the moisture susceptibility of waste cooking oil rejuvenated-asphalt mixture is little" it doesnt sound good.
- reference 9 does not have the year of publication.
- Pls change the word " Saturation" in Table 11 to Saturates.
- Unit of "mPaï¹’s" should be mPa.s without space (line 313 and Table 6.
Reviewer 2 Report
Comments to authors
This study reviewed the literature regarding the possibilities of using waste cooking oil in asphaltic materials. The technical writing and presentation of the manuscript are not up to the mark. Hence, the manuscript requires major modification and justification before publishing as comments are provided below
o Authors must take into account that the “Abstract” is one of the most important parts of scientific papers. However, the abstract here is not well-written. The abstract must be improved by deleting unnecessary info and providing more findings, contributions, and novelty as well as suggestions for the following studies.
o Why the intro has 0 as the heading?
o The intro is not well-written and needs modification. For example, providing an overview of existing materials and narrowing down with more in-depth discussion on their impacts with comparison and justification.
o Why there are no references for several statements such as Lines 50 to 53, 59 to 69, 195 to 201, 251 to 257, etc.?
o A more critical introduction and comparison between previous studies can considerably enhance the quality of sections 1 and 2. For example, the author can compare and discuss (in detail) the findings of studies with the application of a similar proportion of waste oils.
o References for tables and figures should be provided.
o The author can also improve the paper by providing the methods that have been utilized (in different studies) to perform tests and evaluate the materials' performance.
o Lines 303 to 309 are only one sentence and it is not clear.
o Lines 326 to 333 need a better explanation and more info should be provided for example what are the oils' differences?
o Can the author justify the statement outlined in lines 454 to 456 “The microcosmic appearance of rejuvenated asphalt is apparently no longer homogeneous, which can improve the performance of asphalt"?
o The author should add a discussion section to compare all the studies and provide some suggestions.
o The conclusion is very general.
Reviewer 3 Report
I believe that this article can influence researchers in other fields of science and industry, while it needs some improvements. Here are some suggestions for improvement:
Authors must read the information regarding the preparation of the manuscript https://www.mdpi.com/journal/materials/instructions#preparation
For example: "The abstract should be a total of about 200 words maximum."
The Introduction contains too much information of little scientific interest and information that is common knowledge, e.g., lines 46-68. What is missing is a detailed literature review of various rejuvenators in particular rejuvenators using used oils and their derivatives that enable temperature reduction, since the solution described in the article also enables temperature reduction. It would be worthwhile for the authors to include the following publications:
Processing of bitumens modified by a bio-oil-derived polyurethane
Influence of waste engine oil addition on the properties of zeolite-foamed asphalt.
Influence of six rejuvenators on the performance properties of reclaimed asphalt pavement (rap) binder and 100% recycled asphalt mixtures
In order for the review of the state of research made by the author to be credible, it is worth making a list indicating what parameters were taken into account in the selection of the analyzed publications, e.g. keywords, names of journals, year of publication, etc.
It is unclear what was the basis for the development of the figures and tables. The lack of literature reference suggests that the figures and tables were created based on the authors' research results. It is also not understood why the authors presented "Segregation test result of blended asphalt" in the figure?
In conclusion, the authors should point out the pros and cons of using WCO in asphalts. Equally important is the development of a method of procedure describing how to classify WCO in the context of suitability for asphalts, as well as an indication of tests to be performed on asphalts and mix asphalts with WCO. Perhaps, on the basis of a literature review, it will be possible to identify WCO limits or critical test results. Only after these additions will the article have significant scientific value.
Round 2
Reviewer 1 Report
All comments were addressed, no further comments
Thank you
Reviewer 2 Report
Comments to authors
This study reviewed the literature regarding the possibilities of using waste cooking oil in asphaltic materials. The technical writing and presentation of the manuscript are still not satisfactory. It is recommended to rewrite the paper and resubmit it. Some comments are provided below:
o What do the authors mean by stating "The technical standard of waste cooking oil applied in 30 asphalt will be formed." in the abstract? Based on their claim the waste cooking oil has negative impacts on both modified asphalt and modified asphalt mixture based on reduction in rutting and fatigue resistance (lines 21-22).
o The intro still needs more justification for example lines 51 and 52, the authors must provide a reference that cooking oil may not produce secondary pollutants. Same for lines 60 to 63.
Please explain what this claim (line 54) means? “the function of waste control by waste”
The intro is not well organized and well written. The entire introduction needs to be rewritten.
o Section 2 first paragraph, it is not clear if the authors cited approximately 17 papers or just 2 papers [13, 30]. Please explain why limited discussion is only provided for 4 lines if the authors cited 17 papers. In addition, the sentence should be amended since the grammar is not correct.
Same section line 186 what does it mean? “To achieve good the performance of asphalt binder and asphalt mixture with waste cooking oil”
o Section 3 it is mentioned that fatigue performance improved (lines 282 to 284) while another claim was presented earlier (lines 108 to 112). Such contradiction requires more in-depth justifications.
Section 3 line 320 what does the author mean by the reduction in construction safety? And it is not clear if they Cited 2 papers or 15 papers. Authors need to be consistent with symbols.
o Can the author justify the statement outlined in lines 454 to 456 “The microcosmic appearance of rejuvenated asphalt is apparently no longer homogeneous, which can improve the performance of asphalt"? previous comment. Why did they remove their claim instead of providing justification?
o The author did not add a discussion section to compare all the studies and provide some suggestions.
Round 3
Reviewer 2 Report
The added “discussion section” is well-written and well-organized and considerably elevated the level of the paper. However, the rest of the paper still needs major revision for better understanding as mentioned in the previous review round.
